# Pharmacokinetics and Pharmacodynamics of Butorphanol and Dexmedetomidine after Intranasal Administration in Broiler Chickens (*Gallus gallus domesticus*)

**DOI:** 10.3390/vetsci9050212

**Published:** 2022-04-25

**Authors:** Jin Sha, Kavitha Kongara, Preet Singh, Antony Jacob, Jeyamohan Ponnampalam

**Affiliations:** 1Wild Base, School of Veterinary Science, Massey University, Palmerston North 4472, New Zealand; feiyu1994987@gmail.com; 2Animal Welfare Science and Bioethics Centre, School of Veterinary Science, Massey University, Palmerston North 4472, New Zealand; p.m.singh@massey.ac.nz (P.S.); a.jacob@massey.ac.nz (A.J.); j.ponnampalam@massey.ac.nz (J.P.)

**Keywords:** intranasal, pharmacokinetics, sedation, analgesia, butorphanol, dexmedetomidine, chicken

## Abstract

Butorphanol and dexmedetomidine (DXM) can produce analgesia in birds. Intranasal (IN) route of drug administration is easier, and free of risks such as pain and tissue damage compared with intravenous, intramuscular or subcutaneous routes in bird species, including wild birds. Although previous studies have demonstrated the use of IN route for producing sedation, no studies are available on the pharmacokinetics and pharmacodynamics of IN drugs in birds. This study analyzed the pharmacokinetics and sedative–analgesic efficacy of intranasal butorphanol (2 mg/kg), dexmedetomidine (80 µg/kg) and their combination (butorphanol, 2 mg/kg; DXM, 80 µg/kg) in healthy, male, Ross broiler chickens (*n* = 6/group) aged between 6 and 8 weeks. Maximum plasma concentration (Cmax, *p* = 0.01), area under the plasma concentration-time curve from time zero to 120 min (AUC_0 to 120_, *p* = 0.02) and apparent volume of distribution at steady state (Vss, *p* = 0.02) of DXM were significantly higher than that of DXM co-administered with butorphanol. The mechanical nociceptive thresholds and the sedation scores of DXM group were significantly higher than the baseline value. Dexmedetomidine (80 µg/kg, IN) was effective in chickens, and the drug absorption was more rapid than that of DXM with butorphanol. However, the duration of action of DXM was short. Lower value of Cmax and nociceptive thresholds showed the nonsignificant efficacy of butorphanol at a dose of 2 mg/kg after IN administration in broiler chickens.

## 1. Introduction

Sedation and analgesia may be advantageous in birds to avoid stress, anxiety and struggling during diagnostic investigations, therapeutic manipulations and painful conditions [1,2]. Intramuscular (IM) injections into the pectoral muscles may incur the risk of pain and muscle necrosis [3]. Inappropriate injections into thigh muscles could cause nerve damage, and there is a risk of excretion of the injected drug before its absorption due to activation renal portal system controlled by autonomic nervous system in birds [4]. The absorption of drugs after subcutaneous (SC) injection can be slower as compared to intravenous (IV) administration, resulting in slower onset of action [5]. Previous studies in birds demonstrated that several classes of sedative–analgesics, such as benzodiazepines, opioids and alpha 2 agonists, are effective when administered intranasally [6,7,8]. The highly vascular and large absorptive nasal mucosal surface favor rapid drug uptake [9]. It can be used to produce sedation and analgesia, which is suitable to restrain the bird for diagnostic and therapeutic manipulations [10]. Advantages of the IN route also include higher client satisfaction as it is non-invasive and not painful compared to the IM route [10]. Rapid onset of sedative–analgesic (SA) drug effects is observed following IN administration [6,10].

Butorphanol is a SA opioid which has higher affinity for kappa receptors than µ- opioid receptors in birds [11]. It has been shown to be a more effective analgesic than morphine in chickens [12]. The effects of butorphanol can be augmented by concurrent administration of other SA drugs, such as dexmedetomidine (DXM) [13]. The injectable forms of both drugs can be administered intranasally in birds [7,10]. The recommended sedation and analgesic doses for IN administration range from 1 to 3 mg/kg for butorphanol [10], and 80–100 µg/kg for DXM [7] in birds. Pharmacokinetic (PK) data coupled with data on pharmacodynamics (PD) can provide a comprehensive estimate of drug effects in the given dose and route. No studies are available on PK and analgesic effects of IN butorphanol and DXM in birds, but sedative efficacy studies on IN midazolam and butorphanol in cockatiels are available [6]. Behavior-based sedation scales have been used to evaluate sedation in birds [14] and mechanical nociceptive threshold testing can be used to evaluate the efficacy of analgesics in birds [15].

The aim of the current study was to determine the pharmacokinetics and pharmacodynamics of intranasal butorphanol, dexmedetomidine and their combination in broiler chickens (*Gallus gallus domesticus*). The pharmacodynamic effect was assessed by evaluation of the sedation and analgesia.

## 2. Materials and Methods

The study was approved by the Massey University Animal Ethics Committee (MUAEC, protocol number 18/31, dated 15 June 2018).

### 2.1. Pharmacokinetic Study

#### 2.1.1. Experimental Animals

Eighteen healthy male Ross broiler chickens, aged between 6 and 8 weeks, were randomly selected from a flock of 200 raised at Massey University Poultry Research Unit. The health condition of the birds was screened by a veterinarian who regularly visits the poultry unit. Birds were weighed by the poultry unit personnel the day prior to testing. The mean (±SD) body weight of the chickens was 2.05 (±0.03) kg.

The experiment was conducted over a three-day period. On each day of the trial, the study chickens were transferred to individual cages in a room adjacent to their usual housing 30 min prior to the start of the experiment. The study chickens were always in visual contact with other chickens.

#### 2.1.2. Drug Administration and Blood Sampling

The chickens were randomly divided into three groups of six each. Group 1 (*n* = 6) received IN butorphanol (10 mg/mL, Ilium Butorgesic, Injection Troy Animal Health care, Sydney, AU, at a dose of 2 mg/kg, group 2 (*n* = 6) received IN DXM (0.5 mg/mL, Dexdomitor, Injection, Orion Corporation, Espoo, Finland) at a dose of 80 µg/kg, and group 3 (*n* = 6) received a combination of IN butorphanol (2 mg/kg) and DXM (80 µg/kg). The dose of the test drugs was chosen based on their use in birds in previous studies [7,12].

The chickens were restrained by hand for drug administration. Tuberculin syringes (1 mL) without the needle that replaced by a catheter sleeve (20 G, 5/8”) were used for accurate delivery of the test drugs. The tip of the catheter sleeve was cut about half of its length and the beveled end was introduced into the nostril. The total dose of the test drugs was divided between both nostrils and administered over 3–5 s. A medial metatarsal vein was catheterized (22 G, 5/8” catheter) aseptically, to sample blood (1 mL each time) at 0 (before drug administration), 2, 5, 10, 20, 30, 45, 60, 90 and 120 min after drug administration. The catheters remained in place for the entire blood collection period. The blood samples were kept on ice immediately after collection and centrifuged at 1000 rpm for 10 min. Plasma was harvested and stored at −80 °C until analysis.

#### 2.1.3. Drug Determination in Plasma

The plasma concentrations of the test drugs were analyzed using a liquid chromatography and mass spectrometry (LCMS) method. The Ultra-High-Performance Liquid Chromatography system (Dionex UltiMate 3000 System; Thermo Scientific, Waltham, CA, USA) comprises a vacuum degasser, a tertiary loading pump, a column oven and an autosampler. A 100 × 2.1 mm column with 2.6 μm particle size (Accucore 150 C_18_ Column; Thermo Scientific, Waltham, CA, USA) was applied with an identically packed guard column (Accucore Defender Guard Column; Thermo Scientific, Waltham, CA, USA). A hybrid quadrupole orbitrap mass spectrometer (Q Exactive Focus; Thermo Scientific, Waltham, CA, USA) was used in mass spectrometry detection.

#### 2.1.4. Sample Preparation

For DXM, a 100 μL plasma sample was vortex mixed with 600 μL methanol for 5 min. After that, the mixture was centrifuged at 12,000 rpm for 10 min. The supernatant was pipetted to a clean glass tube and eluted in a phospholipid removal tube (Phree Phospholipid removal; Phenomenex, Torrence, CA, USA). The eluate from the phospholipid removal tube was collected in a clean glass tube and dried in a vacuum evaporator (SpeedVac, Thermo Fisher Scientific, Auckland, New Zealand). The LCMS water (100 μL) was used to reconstitute the dried residue, and 10 μL of the mixture was injected into the LCMS column.

For butorphanol, a 100 μL plasma sample and 400 μL methanol was vortexed mixed for 10 min. Then, the mixture was centrifuged at 12,500 rpm for 10 min. The supernatant was transferred to a phospholipid removal tube (Phree Phospholipid removal; Phenomenex, Torrence, CA, USA). The eluate from the phospholipid removal tube was collected in a clean glass tube and dried in a vacuum evaporator (SpeedVac, Thermo Fisher scientific, Auckland, New Zealand). The methanol (100 μL) was used to reconstitute the dried residue, and the mixture was centrifuged at 12,000 rpm for 5 min. After centrifuging, 10 μL of the mixture was injected into the LCMS column.

#### 2.1.5. Liquid Chromatography and Mass Spectrometry (LCMS) Conditions

For DXM, the mobile phase was composed of 0.1% formic acid in H_2_O and 0.1% formic acid in acetonitrile at a ratio of 90:10. The isocratic flow rate was 0.3 mL/min and the run time was six minutes. The temperature of the column and the capillary tube was 25 °C and 320 °C, respectively. The heated electrospray ionization probe was maintained at 3.30 KV and all analyses were performed in the positive ionization mode. Nitrogen drying gas was used in LCMS. The sheath liquid flow was 30 arbitrary units, the auxiliary gas flow was 5 arbitrary units and the ion sweep gas flow was 1 arbitrary unit.

For butorphanol, the mobile phase was composed of 0.1% formic acid in H_2_O and 0.1% formic acid in acetonitrile at a ratio of 75:25. The isocratic flow rate was 0.3 mL/min and the run time was 10 min. Other LCMS conditions of butorphanol were the same as those of DXM (as described above).

#### 2.1.6. LCMS Validation

The blank plasma was spiked with stock solution of drugs which were the same as that used for the treatment of chickens in the research. The validation for both of the drugs was performed separately. Six standards of DXM for the calibration curve were from 0.25–16.6 ng/mL, and six standards of butorphanol for the calibration curve were from 4.1–83.8 ng/mL. The lower limits of quantification (LLQ) in the mobile phase were measured by the signal-to-noise ratio of 10:1. The intra-day variation was determined at five different concentrations of independently prepared spiked plasma on the same day. The inter-day variation was determined at the concentrations for three consecutive days. Recoveries of DXM and butorphanol were determined by comparing the mean area response of unextracted samples (spiked after extraction) with the area of control standards following the same sample preparation.

#### 2.1.7. Pharmacokinetic Analysis

PKsolver add-in program for excel 2010 was used to analyse concentration-time data with non-compartmental method [16]. The maximum plasma concentration (Cmax, ng/mL) of the drugs and time to reach Cmax (Tmax, min) were determined as direct observation from the plasma drug concentration results. The other parameters such as elimination half-life (t_1/2el_, min), area under the concentration–time curve (AUC, ng·min/mL), mean residence time (MRT, min), apparent volume of distribution at steady-state (Vss, L/kg) and clearance (Cl, L/min/kg) were calculated by the linear trapezoidal rule in the PKsolver add-in program.

### 2.2. Evaluation of Sedation and Analgesia

#### 2.2.1. Experimental Animals and Drug Administration

Twenty-four healthy, male Ross broiler chickens, aged between 6 and 8 weeks, were selected from the same unit as those selected for the pharmacokinetics study. The health condition of the birds was screened by a veterinarian who regularly visits the poultry unit. The mean (±SD) body weight of the chickens was 2.15 kg (±0.08). The chickens were randomly divided into four groups. Group 1 (*n* = 6) received butorphanol (2 mg/kg, IN), group 2 (*n* = 6) received DXM (80 µg/kg, IN), group 3 (*n* = 6) received a combination of butorphanol (2 mg/kg) and DXM (80 µg/kg) and group 4 (*n* = 6) received normal saline (about 1 mL as control). The chickens were restrained by hand for drug administration. Tuberculin syringes (1 mL) were used for accurate delivery of the test drugs. The total dose of the test drugs was divided between both nostrils and administered over 3–5 s.

#### 2.2.2. Sedation Assessment

Sedation was assessed at 10 min prior to, and 5, 10, 20, 30, 45, 60, 90 and 120 min after drug administration. Level of sedation achieved was scored using the subjective scoring method described by Pollock et al (2001) [14].

#### 2.2.3. Mechanical Nociceptive Threshold (Analgesia) Testing

Nociceptive thresholds of the chickens were tested after scoring the sedation at 0 (before drug administration), 5, 10, 20, 30, 45, 60, 90 and 120 min. A handheld algometer (Wagner Instruments, Greenwich, CT, USA) was used; a 2 mm diameter tip of the algometer was pressed against the skin on the dorso-lateral aspect of the proximal half of the metatarsal area on the left leg of all chickens. Behavioral responses such as opening the eyes (if closed due to sedation), stretching of the leg, leg withdrawal, sudden shuffling (moving feet without standing) and body twitching [17] were considered as the end point for threshold recording (Figure 1A,B). The cut-off threshold was 9 N to avoid tissue damage. The baseline nociceptive thresholds were measured three times at approximately 30–45 s interval in each chicken. As there were no significant changes in the baseline values over time, the post-treatment nociceptive thresholds of all groups were measured once in each chicken. The investigator assessing the sedation and analgesia was blinded to the treatment groups.

#### 2.2.4. Statistical Analysis

Normal distribution of plasma concentration data was assessed by the Kolmogorov–Smirnov test. The significant difference between the parameters of butorphanol and butorphanol combined with DXM was analyzed by paired *t*-test in Excel. The parameters of DXM and DXM combined with butorphanol were also analyzed by paired *t*-test for the significant difference. Values were considered significant at *p* < 0.05. The differences of the parameters between butorphanol group and DXM group were not considered because of the different drugs and doses.

Mechanical nociceptive threshold data were tested for normality using a Kolmogorov–Smirnov test. A generalized linear mixed model (multivariate ANOVA) with repeated measures was used to analyse the normally distributed nociceptive thresholds followed by post hoc test (Fisher’s least significant difference comparison). The measurements of nociceptive thresholds were expressed as the mean and standard deviation (SD). The distribution of sedation score data was non-normal and hence, the raw scores were first rank-transformed using ‘proc rank’ procedure in SAS^®^ 9.4. to test differences between-group and between time-points, within a group. A mixed model analysis of the transformed sedation scores was carried out using ‘proc mixed’ procedure in SAS^®^ 9.4. The model included the fixed effects of group, time and their interaction, and random effects of chickens. Sedation scores were presented in Table 1 as least square mean ± standard error (LSM ± SE), in the original scale. Values were considered significant at *p* < 0.05.

## 3. Results

### 3.1. LCMS Validation and Pharmacokinetic Analysis

The inter-day variation of DXM in plasma ranged 4.5–7.5% and the intra-day variation was below 5%. The correlation coefficient was 0.9983 for the standard curves of DXM. The recovery of DXM ranged 65–78% and the LLQ was 0.25 ng/mL. The inter-day variation of butorphanol in plasma ranged 3.9–10.2% and the intra-day variation ranged 1.8–4.4%. The correlation coefficient was 1 for the standard curves of butorphanol. The recovery of butorphanol ranged 82−85% and the LLQ was 4 ng/mL.

The chromatograms of butorphanol at concentrations of 8.3 and 16.6 ng/mL, including blank plasma, and at 20 and 60 min after intranasal administration in chickens are shown in Figure 2. Figure 3 shows the chromatograms of DXM at concentrations of 4 and 8 ng/mL, including blank plasma, and at 2 and 45 min after intranasal administration in chickens.

The pharmacokinetic data are shown in Table 2. Both the Cmax and AUC_0 to 120_ values of DXM were significantly higher than DXM with butorphanol (*p* = 0.01, 0.02, respectively). The MRT value of DXM was lower than that of DXM with butorphanol.

The semi log plot of concentration–time profiles of butorphanol and butorphanol combined with DXM in chickens after intranasal administration are shown in Figure 4, and that of DXM and DXM combined with butorphanol are shown in Figure 5.

### 3.2. Sedation Score

The sedation scores did not show significant difference between butorphanol and saline groups. The sedation scores of DXM group were significantly higher than that of saline group at 5, 10, 20, 30, 45, 60 and 90 min (*p* = 0.02, 0.004, 0.003, 0.01, 0.002, 0.01, 0.01, respectively). There was no significant difference in the sedation scores between the butorphanol and DXM combination and saline groups (Table 1).

Within groups, the sedation scores at 90 min were significantly higher than that of baseline in butorphanol group (*p* = 0.02). The sedation scores of DXM group at 5, 10, 20, 30, 45, 60, 90 and 120 min were significantly higher than that of the baseline (*p* = 0.002, 0.002, 0.002, 0.002, 0.001, 0.002, 0.002, 0.005, respectively). The sedation scores of the combination of butorphanol and DXM group at 10, 20, 30, 45, 60, 90 and 120 min were significantly higher than that at the baseline (*p* = 0.02, 0.02, 0.01, 0.02, 0.006, 0.02, 0.02, respectively).

### 3.3. Mechanical Nociceptive Thresholds

There were several within-group differences as compared to their respective baseline values (Table 3). The nociceptive thresholds of DXM group were significantly higher than their baseline values at all time-points after the drug administration. The mean (±SD) threshold values were lowest at 5 min, reached a peak at 30 min and started dropping after 90 min post-drug administration. No significant differences were found between the baseline and post-treatment thresholds in the butorphanol group. However, the thresholds at 30 and 45 min were significantly lower than that at 120 min in this group (*p* = 0.03 and 0.03, respectively). The nociceptive thresholds of the combination group were significantly higher than the baseline value at 30 (*p* = 0.04), 45 (*p* = 0.03) and 120 (*p* = 0.005) minutes after the drug administration but the trend was fluctuant (Figure 6).

The baseline thresholds did not differ significantly between the treatment groups. Post-treatment thresholds of the DXM group were significantly higher than the saline group at all time points and except at 120 min in the butorphanol group (Table 3 and Figure 6). There was no significant difference in the post-treatment thresholds between butorphanol and saline groups except at 120 min. The nociceptive thresholds were significantly different between the saline group and the combination group (*p* = 0.01).

## 4. Discussion

Sedative–analgesic drugs such as butorphanol or DXM were shown to be effective in birds [18,19]. Although parenteral route (IV, IM or SC) has been the common method for drug administration in birds, the IN route is preferred [10,20] due to the drawbacks associated with the former.

Previous studies have shown that the combination of midazolam and butorphanol after IN administration produced a better sedative effect in cockatiels (*Nymphicus hollandicus*) than IN midazolam alone [6], and a combination of midazolam and DXM after IN administration caused a more obvious sedative effect than IN midazolam alone in pigeons (*Columba livia*) [7]. In mice, the combination of butorphanol and DXM could produce better sedation and antinociception resulting from synergistic interaction between the opioid and alpha2 receptors [21]. No studies are available on the effect of IN burorphanol or DXM or a combination of DXM and butorphanol in birds.

This is the first study reporting the pharmacokinetics of butorphanol and DXM in any avian species after intranasal administration. The Cmax achieved after IN administration of butorphanol at 2 mg/kg in this study was much lower than the Cmax after its IV administration in broiler chickens at the same dose [22]. The lower plasma concentration profile of butorphanol in the current study could be due to absorption of some of the total dose through the mouth as the total volume of drug administered could be more than the volume of the nasal cavity. Also, the secondary palate is incomplete in chickens [23] and drugs can pass into the oral cavity through the choanal cleft. Since oral bioavailability of butorphanol is poor [19], much of the drug absorbed orally must have gone through the extensive hepatic metabolism resulting in lower plasma concentrations as compared to other routes of administration.

Dexmedetomidine, when given IN, was rapidly absorbed to achieve a Cmax of 2.31 ± 0.19 ng/mL, which was significantly higher than DXM administered in combination with butorphanol (1.43 ± 0.20 ng/mL). Similarly, AUC_0 to 120_ was significantly higher in DXM alone group as compared to DXM with the butorphanol group. In contrast, the Cmax and AUC_0 to 120_ of butorphanol were higher in the combination group as compared to the butorphanol alone group. However, these differences did not achieve significance. The sample size of the treatment groups (*n* = 6 per group) in this preliminary study was based on a similar study that used intranasal midazolam singly (*n* = 6 per group) or combined DXM (*n* = 6 per group) in pigeons [7]. Differences in cardiovascular variables and sedation scores between the treatment groups were found in that study. In the current study, a post-hoc power analysis pertaining to the pharmacokinetic data of butorphanol with DXM and butorphanol alone groups showed a low power (65–70%) to detect between-group differences. Increasing the sample size could increase the power to find significant differences between these groups. Small sample size is one of the limitations of the current study.

Both butorphanol and DXM are metabolized in mammals by hepatic hydroxylation and glucuronidation [24,25], and a study has reported synergism in their effect in mice [21]. There are no reports of pharmacokinetic interactions of butorphanol and DXM in any avian species. In this study, metabolic study was not undertaken, thus it cannot be confirmed whether either of these drugs influence their pharmacokinetics when given in combination. Intranasal administration of drugs is designed to bypass hepatic first pass metabolism. Butorphanol at a low dose (given in current study) could achieve analgesic plasma concentrations after IV administration [25]. In the current study, it was not achieved which could be due to variable absorption of the drug through the oral route. The variable absorption of drugs could also result in higher standard deviation for calculation of pharmacokinetic parameters.

One of the limitations of this study was it did not investigate dose–response curve both for pharmacokinetic and pharmacodynamic experiments. The dose–response curve would help us in evaluating the best dosing regimen to provide analgesia for these drugs alone or in combination. In the pharmacodynamics study, there was no significant difference in the nociceptive thresholds and sedation scores before and after treatment in the butorphanol group overtime, which indicated that there was no significant analgesic and sedative efficacy of IN butorphanol (2 mg/kg) in chickens. The nociceptive thresholds of DXM alone group were significantly higher at all time points compared to its baseline values and saline group. The maximum drug effect, in terms of nociceptive thresholds, of DXM reached at 30 min after the drug administration, but the Tmax of DXM was about 25 min. This could be due to a hysteresis of the drug effect or threshold recording after 25 min.

The mechanical nociceptive thresholds of the combination of butorphanol and DXM were significantly lower than that of DXM alone. This could be due to the dilution of DXM concentration when mixed with butorphanol solution as only DXM produced significant analgesia compared to nonsignificant analgesic effects of butorphanol in this study. In addition, the larger volume of the solution of combination of butorphanol and DXM (around 0.72 mL for an around 2 kg bird) might cause a leak of the drugs as compared to that of butorphanol and DXM alone.

Although the nociceptive thresholds of the combination group showed significant increases compared to the saline group at a few time points, this rise in nociceptive thresholds was not consistent. The fluctuations in thresholds could be due to the weak effect of diluted DXM or due to the interaction of the two drugs. Post-treatment sedation scores of the combination group significantly differed compared to the pre-treatment values but there was no significant difference between the combination and saline groups. This indicates a weak and nonsignificant sedative efficacy of the combination.

Sedation may be required for common clinical procedures (radiography, blood collection, physical examination, etc.) to reduce vocalization and stress response caused by manual restraint [10]. In the present study, DXM, administered singly at a dose of 80 µg/kg, had both sedative and analgesic effects from 5 to 120 min after the drug administration. Absence of analgesia and sedation observed in the butorphanol group could be due to a low Cmax of the drug at a dose of 2 mg/kg in the current study.

The doses of butorphanol and DXM used in this study were 2 mg/kg and 80 µg/kg, respectively. Intravenous butorphanol, and intranasal DXM combined with midazolam at these doses were shown to be effective in other studies in birds [7,12]. However, the results of this study indicated that butorphanol at a dose of 2 mg/kg administered by the IN route was not high enough to provoke satisfactory sedation and anti-nociception in chickens. Although a higher dose of IN butorphanol (3 mg/kg) combined with midazolam has been reported to be effective in producing rapid sedation in cockatiels [6], none of the previous studies has shown the effect of butorphanol alone after IN administration in birds.

The data generated from the present study can be used for extrapolation of dosing regimen for both the drugs in other bird species in which it is impossible to conduct such experiments. The extrapolation of data from one bird species to another bird species is more accurate as compared to extrapolation from mammals [26], but these data extrapolations should be made with caution.

## 5. Conclusions

Dexmedetomidine (80 µg/kg) after IN administration was effective in producing sedation and antinociception in broiler chickens, and the drug absorption was more rapid than that of DXM co-administered with butorphanol. However, the duration of the efficacy of DXM was short. Although the dosing interval of DXM appears to be about 2 h in chickens, it cannot be concluded as this preliminary study did not conduct the pharmacokinetic–pharmacodynamic modelling of the drug. The efficacy of butorphanol and butorphanol combined with DXM after IN administration need to be studied further probably with a dose higher than 2 mg/kg.

## Figures and Tables

**Figure 1 vetsci-09-00212-f001:**
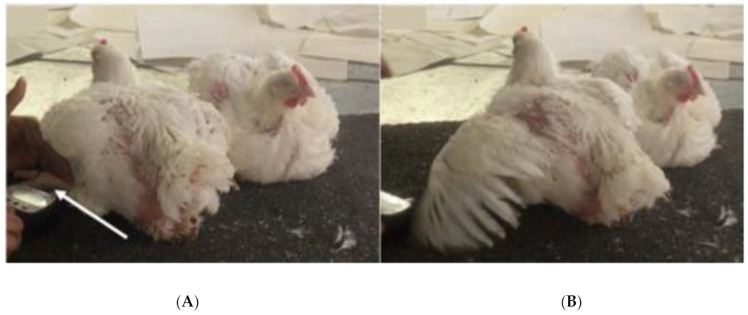
(**A**) The condition of the broiler chickens after administration of dexmedetomidine (80 μg/kg) intranasally (the tip of the algometer probe on the chicken’s leg indicated by the white arrow); (**B**) one of the behavioral responses of the chickens to the algometer used (the end point for threshold recording).

**Figure 2 vetsci-09-00212-f002:**
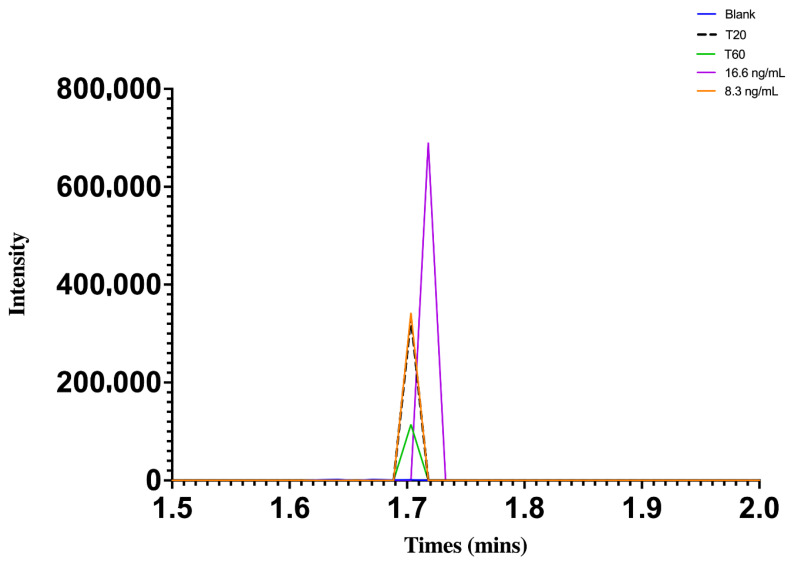
Chromatograph showing butorphanol peak at concentrations of 8.3 and 16.6 ng/mL, after 20 and 60 min of intranasal administration at the dose of 2 mg/kg in broiler chickens.

**Figure 3 vetsci-09-00212-f003:**
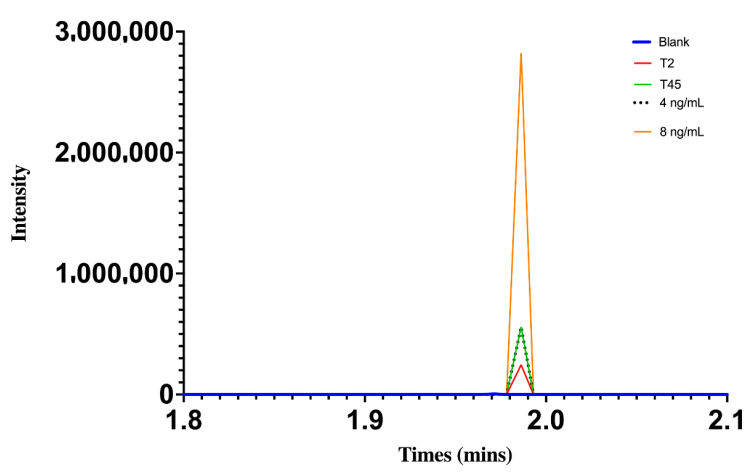
Chromatograph showing DXM peak at concentrations of 4 and 8 ng/mL, after 2 and 45 min of intranasal administration at the dose of 80 μg/kg in broiler chickens.

**Figure 4 vetsci-09-00212-f004:**
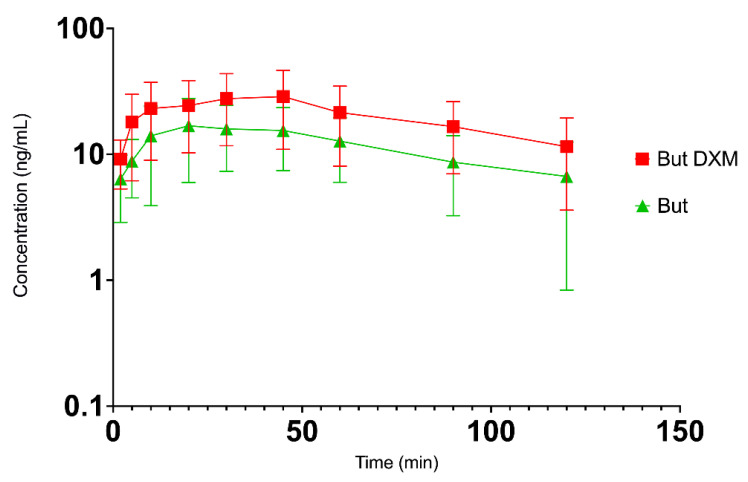
Semi-log plot of mean (±SD) plasma concentration-time profiles of butorphanol (But, green line) and butorphanol with dexmedetomidine (But DXM, red line) after intranasal administration in broiler chickens at 2 mg/kg and 80 µg/kg, respectively (*n* = 6/group).

**Figure 5 vetsci-09-00212-f005:**
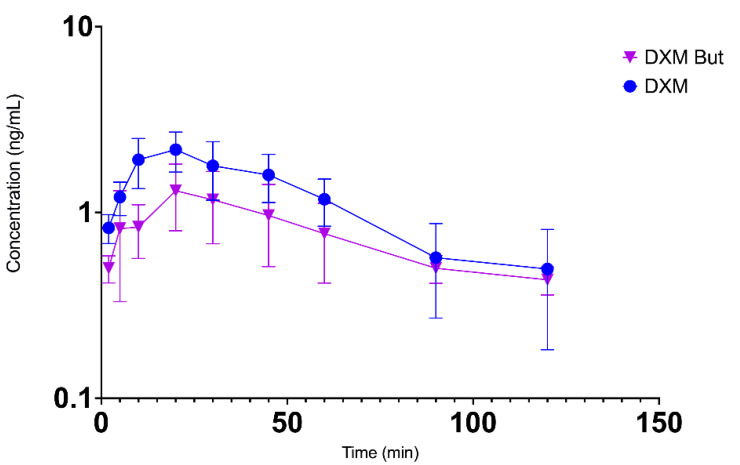
Semi-log plot of mean (±SD) plasma concentration-time profiles of dexmedetomidine (DXM; blue line) and dexmedetomidine with butorphanol (DXM But; pink line) after intranasal administration in broiler chickens at 80 µg/kg and 2 mg/kg, respectively (*n* = 6/group).

**Figure 6 vetsci-09-00212-f006:**
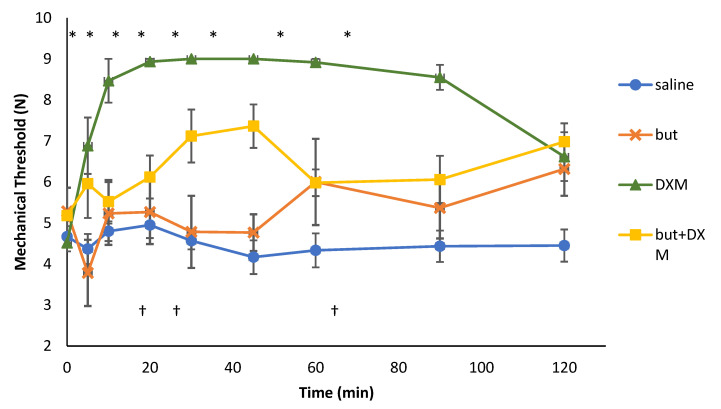
Mechanical nociceptive thresholds (Newton, mean ± SD) of broiler chickens (*n* = 6/group) following intranasal administration of saline (about 1 mL), butorphanol (But, 2 mg/kg), dexmedetomidine (DXM, 80 µg/kg) or a combination of both drugs in same doses. * denotes significant difference from baseline value for DXM, ^†^ denotes significant difference from baseline value for the combination of butorphanol and DXM (*p* < 0.05). but, butorphanol; DXM, dexmedetomidine; but + DXM, the combination of butorphanol and dexmedetomidine.

**Table 1 vetsci-09-00212-t001:** Sedation scores (least square mean ± standard error) of broiler chickens (*n* = 6 per group) after intranasal administration of saline (about 1 mL), butorphanol (But, 2 mg/kg), dexmedetomidine (DXM, 80 µg/kg) or the combination of butorphanol (But, 2 mg/kg) and dexmedetomidine (DXM, 80 µg/kg).

Time (min)	Sedation Score ^§^ (Least Square Mean ^†^ ± Standard Error) in Different Groups			
	Saline	But	DXM	But + DXM
0	0 ± 0.22	0 ± 0.27	0 ± 0.22	0 ± 0.26
5	0.50 ^a^ ± 0.22	0.50 ^a^ ± 0.27	1.50 *^b^ ± 0.22	0.33 ^a^ ± 0.26
10	0.17 ^a^ ± 0.22	0.50 ^ac^ ± 0.27	1.67 *^b^ ± 0.22	0.83 ^c^ ± 0.26
20	0.17 ^a^ ± 0.22	0.67 ^ac^ ± 0.27	2.17 *^b^ ± 0.22	0.83 ^c^ ± 0.26
30	0.50 ^a^ ± 0.22	0.50 ^a^ ± 0.27	2.17 *^b^ ± 0.22	0.83 ^a^ ± 0.26
45	0.17 ^a^ ± 0.22	0.50 ^ac^ ± 0.27	2.17 *^b^ ± 0.22	0.83 ^c^ ± 0.26
60	0.50 ^a^ ± 0.22	0.50 ^a^ ± 0.27	2.17 *^b^ ± 0.22	1.00 ^a^ ± 0.26
90	0.33 ^a^ ± 0.22	0.83 ^a^ ± 0.27	1.33 *^b^ ± 0.22	0.83 ^ab^ ± 0.26
120	0.33 ^a^ ± 0.22	0.67 ^a^ ± 0.27	0.83 ^a^ ± 0.22	0.83 ^a^ ± 0.26

**^§^** Sedation scores were not distributed normally. Hence, they were rank-transformed and analyses using a mixed model analysis for testing the significant differences between groups and time-points. The least square mean presented in this table are in original scale. ^†^ Least square mean values with an asterisk (*) differ significantly (*p* < 0.01) with respective baseline (time 0 min) value. Least square mean values, within each time-point, with at least one common alphabet as superscript do not differ significantly (*p* > 0.05).

**Table 2 vetsci-09-00212-t002:** Pharmacokinetic parameters (mean ± SD) of butorphanol (But, 2 mg/kg; *n* = 6/group), dexmedetomidine (DXM, 80 µg/kg; *n* = 6/group), and their combination at same doses (*n* = 6/group) in broiler chickens, analyzed by a non-compartmental model.

PK Parameter (Unit)	But	But with DXM	DXM	DXM with But
T_1/2 elim_ (min)	69.81 ± 20.26	77.07 ± 13.96	55.08 ± 11.62	152.23 ± 74.87
Tmax (min)	30.00 ± 5.00	29.17 ± 8.21	25.83 ± 4.17	20.83 ± 3.75
Cmax (ng/mL)	23.05 ± 6.21	34.54 ± 6.92	2.31 ± 0.19 *	1.43 ± 0.20
AUC_0 to 120_ (ng min/mL)	1626.58 ± 389.94	2443.78 ± 481.71	144.53 ± 13.40 *	92.94 ± 12.05
AUC_0 to inf_ (ng min/mL)	2637.03 ± 864.16	3716.11 ± 776.85	193.46 ± 29.79	192.69 ± 49.68
MRT (min)	110.40 ± 29.11	116.48 ± 17.41	82.76 ± 16.65	218.48 ± 104.46
Vss/F (L/kg)	107.80 ± 31.91	68.66 ± 13.91	33.19 ± 5.84 *	76.11 ± 16.29
Cl/F (L/min/kg)	2.55 ± 0.77	1.27 ± 0.2	0.95 ± 013	1.03 ± 0.7

* Denotes significant difference from DXM with butorphanol value for DXM (*p* < 0.05). T_1/2 elim_, elimination half-life; Cmax, maximum plasma concentration; Tmax, time of Cmax; AUC_0 to 120_, area under the plasma concentration-time curve from time zero to 120 min; AUC_0 to inf_, area under the concentration time curve from time zero to infinity; MRT, mean resident time; Vss/F, apparent volume of distribution at steady state after non-intravenous administration; Cl/F, apparent total clearance of the drug from plasma after non-intravenous administration; But, butorphanol; DEX, dexmedetomidine; But with DXM, butorphanol combined with DXM; DXM with butorphanol, DXM combined with butorphanol.

**Table 3 vetsci-09-00212-t003:** Mean (±SD) mechanical nociceptive thresholds (Newtons, N) of broiler chickens after intranasal administration of saline (about 1 mL; *n* = 6), butorphanol (But, 2 mg/kg; *n* = 6), dexmedetomidine (DXM, 80 µg/kg; *n* = 6) or both drugs at same doses (*n* = 6).

Time (Minute)	Mechanical Nociceptive Threshold (N; Mean ± SD)			
	Saline	But ^d^	DXM ^ac^	But + DXM ^b^
0 (baseline)	4.67 ± 0.26	5.28 ± 0.58	4.52 ± 0.21	5.18 ± 0.13
5	4.37 ± 0.37	3.78 ± 0.81	6.88 ± 0.69 *	5.96 ± 0.84
10	4.8 ± 0.24	5.23 ± 0.77	8.47 ± 0.53 *	5.52 ± 0.53
20	4.95 ± 0.32	5.27 ± 0.78	8.93 ± 0.07 *	6.12 ± 0.52
30	4.57 ± 0.21	4.78 ± 0.88	9 ± 0 *	7.12 ± 0.65 ^†^
45	4.17 ± 0.41	4.77 ± 0.45	9 ± 0 *	7.36 ± 0.53 ^†^
60	4.33 ± 0.41	6 ± 1.05	8.92 ± 0.08 *	5.98 ± 0.32
90	4.43 ± 0.38	5.37 ± 0.74	8.55 ± 0.31 *	6.06 ± 0.58
120	4.45 ± 0.39	6.32 ± 0.65	6.62 ± 0.59 *	6.98 ± 0.45 ^†^

* denotes significant difference from baseline value for the time point in DXM group, ^†^ denotes significant difference from baseline value for the time point in the combination of butorphanol and DXM group. ^a^ denotes significant difference from saline group for DXM group. ^b^ denotes significant difference from saline group for the combination of butorphanol and DXM group. ^c^ denotes significant difference from DXM group for the combination of butorphanol and DXM group. ^d^ denotes significant difference from butorphanol group for the combination of butorphanol and DXM group (*p* < 0.05).

## Data Availability

The data that support the findings of this study are available from the corresponding author upon reasonable request.

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
