# Peer review of "Pharmacokinetics and Pharmacodynamics of Butorphanol and Dexmedetomidine after Intranasal Administration in Broiler Chickens (*Gallus gallus domesticus*)"

_vetsci, 2022, doi:10.3390/vetsci9050212_

Round 1

Reviewer 1 Report

The manuscript has been revised according to the suggestions of the reviewer's remarks and the manuscript sounds well from scientific point of view. It can be accepted for publication.

Some editorial remarks should be taken into account:

Line 125: delete the number "6", leave the text "six".

Line 165: Delete "Figure 1" because it refers tuberculin syringe. As a result, re-number the other pictures/figures. Fig. 2 should be Fig. 1.

Table 3: The meaning of a, b and c for statistically significant differences should be explained under the table.

Author Response

Thank you for your kind review. Please find my responses in the file attached here.

Some editorial remarks should be taken into account:

Line 125: delete the number "6", leave the text "six".

Deleted.

Line 165: Delete "Figure 1" because it refers tuberculin syringe. As a result, re-number the other pictures/figures. Fig. 2 should be Fig. 1.

Renumbered.

Table 3: The meaning of a, b and c for statistically significant differences should be explained under the table.

There is an explanation under the table as:

Least square mean values with an asterisk (*) differ significantly (P <0.01) with respective baseline (time 0 min) value. Least square mean values, within each time-point, with at least one common alphabet as superscript do not differ significantly (P >0.05).

Reviewer 2 Report

Authors did not respond to my comments. This file attached is a response to other review. Based on this answer, I have not more comments.

Author Response

thank you

Reviewer 3 Report

Dear authors:

After Reading the manuscript entitle “Pharmacokinetics and Pharmacodynamics of butorfanol and dexmedetomidine after intranasal administration in Broiler Chickens (Gallus gallus domesticus)”, I consider that the topic of the paper could be interesting. However, it presents important errors in the development of the study, as well as in the writing and structure of the manuscript.

Major comments:

The study is carried out in Broilers Chickens. Are the drugs used in this study  authorized by your country's Drug Agency? I would like you to clarify if the chickens are domestic or production chickens (what to eat later).

The number of animals per group is too low to obtain adequate results and get a quality and reference manuscript. I suggest the inclusion of a greater number of animals per group. Given that authors are aware of this important limitation, I invite the authors to extend the study and present the results again.

In addition, I consider it is very important to include the metabolic study.

I recommend the authors include in the section “experimental Animals” the climatic conditions, feeding and possible fasting of the animals.

Line 84: Are there specific doses for Broilers Chicken?

On the other hand, and in relation to “sample preparation”, “LCMS conditions” and “Pharmacokinetics analysis”, I think it would be interesting and very important to assess the publication of this article develop a method that allows the simultaneous analysis of samples with butorphanol and dexmedetomidine.

With reference to method validation, I miss the internal standard used. As well as the calibration lines.

Line 165: I can`t find Figure 1.

To improve the quality of the manuscript and focusing on the title and objective of it, I consider that figures 2a, 2b, 3a, 3b should be eliminated or put as Annex.

In the Chromatograms (Figure 4 y 5), I consider that the chromatograms do not correspond to the legend. I invite the authors to improve the scale to be able to see the peaks better.

Figure 6 and 7: could you explain figures in more detail.

Line 254: I don't see the introduction of the table in the text. In table 2, I don't understand the presence of the two columns of butorphanol + Dexmedetomidine.

Line 341 to 360: this paragraph would correspond to the introductory section. clarifying several of the questions that readers may initially ask.

Minor comments:

Line 40: specify several classes of sedative-analgesic drug.

Line 45: What drugs are you referring to?

Line 50: what does SA mean?

Line 53: They should try to update the bibliographical references 7 and 10.

Line 53: Unify the use of abbreviations. Abbreviations alternate with the full word. What tarnishes good writing. Review the entire manuscript.

Line 60: In relation to the aim of the study, for what purposes do you want to study pharmacokinetics of butorphanol and desmedetomidine?

Line 79: remove n=18

Line 94 and 108: decide between g or rpm. Review the entire manuscript.

Line 184 and figure title: choose between uppercase or lowercase

Line 443: Throughout the manuscript in some paragraphs it speaks of birds and in another of broilers. please clarify this fact. For example in conclusions it refers to birds but in the tittle it refers to Broiler Chicken.

Correct minor errors: line 125, 130, 174, 192, 252, ….

Author Response

Thank you for reviewing our manuscript. Our responses to your questions/comments have been uploaded below.

Reviewer 4 Report

The aim of the manuscript was to determine the pharmacokinetics and pharmacodynamics of intranasal butorphanol, dexmedetomidine, and their combination in broiler chickens.  A non-compartmental analysis was used to determine the pharmacokinetic parameters and the pharmacodynamic effect was assessed by evaluation of the sedation and analgesia.

The structure and concise wording of the manuscript are adequate, which greatly facilitates its understanding by potential readers. On the other hand, tables and figures in the manuscript are adequate.

The Authors are kindly asked to consider including the sample size as a limitation of the study in the Discussion. The sample size could explain that there are no significant differences in the comparisons.

Author Response

Thank you for your kind review and suggestions to improve our manuscript. I've included in the discussion (lines 392 -393) that a small sample size is one of the limitations of the current study.

Round 2

Reviewer 3 Report

Dear authors,

Thank you for the clarifications provided regarding the questions raised. After revising the manuscript again, I would like to insist on the following changes with the aim of improving the quality of the manuscript:

Enhance the scales of the chromatograms to make them look better. It is important even if they want to prioritize the pharmacokinetic approach.

Eliminate figures 2A and 2B, I do not think they improve the understanding of the study. In any case, if the authors consider them to be of vital importance attach them as an Annex.

I reiterate the need to move the paragraphs from lines 356-373 to the introduction section. By removing these paragraphs from the discussion section, I encourage authors to make a two-line summary of these paragraphs as an introduction to the next paragraph in discussion.

And finally, I would appreciate it if you personally explain the data in Table 2 to me. I managed to understand the difference between DXM+But and But+DXM.

How is the medication administration?

DXM+But: DXM is administered first and after a while, But?

Pero+DXM: But is DXM administered first and after a while?

Thank you very much for your time and congratulations.

Author Response

Thank you for your kind review; please find our response in the attached file.
